# Forest and Arborescent Scrub Habitats of Special Interest for SCIs in Central Spain

**Ana Cano-Ortiz** [1], **Carmelo M. Musarella** [1,2], **Jose C. Piñar Fuentes** [1], **Ricardo Quinto Canas** [3], **Carlos J. Pinto Gomes** [4], **Giovanni Spampinato** [2], **Jehad Mahmoud Hussein Ighbareyeh** [5], **Sara del Río** [6] and **Eusebio Cano** [1,*]

1. Departamento Biología Animal, Vegetal y Ecología Botánica, Universidad de Jaén, 23071 Jaén, Spain; anacanor@hotmail.com (A.C.-O.); carmelo.musarella@unirc.it (C.M.M.); jpinar@ujaen.es (J.C.P.F.)
2. Dipartimento di AGRARIA, Università "Mediterranea" di Reggio Calabria, Località Feo di Vito, 89122 Reggio Calabria, Italy; gspampinato@unirc.it
3. CCMAR—Centro de Ciências do Mar, University of Algarve, 8005-410 Faro, Portugal; rqcanas@gmail.com
4. Departamento de Paisagem, Ambiente e Ordenamento/Instituto de Ciências Agrárias e Ambientais Mediterrânicas (ICAAM), Universidade de Évora, 7000-365 Évora, Portugal; cpgomes@uevora.pt
5. Department of Plant Production and Protection, Faculty of Agriculture, Al-Quds Open University, Hebron Branch, Abu Ktellah Street, Hebron 770-773, Palestine; jehadighbareyeh@hotmail.com
6. Department of Biodiversity and Environmental Management (Botany), Faculty of Biological and Environmental Sciences, University of León, Campus de Vegazana s/n, E-24071 León, Spain; sara.delrio@unileon.es
* Correspondence: ecano@ujaen.es

**Abstract:** The habitat of the several territories in Ciudad Real (Castilla-La Mancha, Spain) are studued through the and mapping (scale 1:10.000) and vegetation analysis. The distribution and surface of the habitat presents in the Sites of Community Interest (SCIs), as well as pressures, threats, trends, and state of conservation are described. These site contributes significantly to the maintenance or restoration at a favourable conservation status of a natural habitat type or of a species of community intesess.These specially protected areas are part of the Natura 2000 network. We discuss the diversity of forest habitats characterized by species of the genus *Quercus* L., focusing only on the plant communities in the Habitats Directive 92/43/EEC of 1992, regarding the conservation of fauna and flora and habitats of interest owing to their endemic or rare character. Habitats and species must be studied in combination to ensure the maximum reliability of the results. We concentrate on habitats with low representation in the territory as a consequence of their rarity or endemicity. We study the following habitats of special interest: 9230—Mediterranean-Ibero-Atlantic and Galaico-Portuguese oak woods of *Quercus robur* and *Quercus pyrenaica*; 9240—Iberian oaks of *Quercus faginea* and *Quercus canariensis*; 9320—Thermomediterranean forests of *Olea* and *Ceratonia* (Iberian Peninsula, Balearic and Canary Islands); 9540—Mediterranean pine forests of endemic *Pinus pinaster* (*Pinus pinaster* subsp. *acutisquama*); 9560—Endemic forests with *Juniperus* spp.; 5210. Arborescent scrub with *Juniperus* spp.

**Keywords:** habitat; mapping; association; vegetation; conservation

## 1. Introduction

The habitat mapping has applications in land planning and management and is a necessary tool in drawing up conservation plans of nature and biodiversity. Forest vegetation, as well as that of other natural and semi-natural habitats, reflects the ecological conditions that occur in a given area, as well as changes in these conditions environmental and human influences. A good understanding of the conservation status and distribution of habitats is essential for planning long-term decisions.

We study the EU forest habitats through map at a scale of 1:10,000 for five sites of community interest (SCIs). The habitats are dominated by species in the genus *Quercus* L. Four of the five sites of community interest (SCIs) were characterized by their bioclimatic,

soil, floristic, and syntaxonomical similarities; hence, some plant associations are repeated in certain areas. The territories in the study all have predominantly siliceous substratess, except for the Lagunas de Ruidera SCI, which is calcareous type. The work areas are located in the central Iberian Peninsula and comprise a series of mountain massifs with a north-west to south-west orientation; this allows areas of low-pressure from the Atlantic to enter the valleys, thereby increasing precipitation and favoring the presence of woodlands of *Quercus pyrenaica* Willd., *Quercus faginea* subsp. *broteroi* (Cout.) A. Camus, *Quercus canariensis* Willd., and *Quercus marianica* C. Vicioso. These sites have a high rate of endemic species and priority EU habitats [1,2], although on more open sites with less rainfall, the dominant woodland is *Quercus rotundifolia* Lam. All these woodlands are found in the form of a "dehesa" or wooded pasture, 6310—Dehesas with evergreen *Quercus* spp. a vegetation structure that is unique to Europe and of vital importance for a wide range of livestock: cows, sheep, goats, and pigs, among others. Holm oak woodlands are particularly vital for pig farming.

In addition to the various types of *Quercus* woodlands, human activity throughout history has led to the establishment of plant communities result from regressive dynamics of the vegetation. This is the case of the copses of *Quercus coccifera* L. with *Juniperus oxycedrus* L. [3], *Arbutus unedo* L. stands, scrublands of *Erica australis* L. and *Erica umbellata* Loefl ex L., and a wide variety of rock rose, broom, and thorny shrubs [4,5]. This work is the result of a joint project between the company GEACAM and the University of Jaén, under a scientific collaboration agreement to study the vegetation, habitat types, and distribution of endangered flora in spaces in the Natura 2000 network in the province of Ciudad Real, Castilla-La Mancha. Its aims are the following: (1) to carry out a detailed digital vegetation map of the Natura 2000 network sites, with a particular focus on the oak habitat in Annex I of Directive 92/43/EC [6], and habitats with special protection measures in Castilla-La Mancha (Law 9/1999 and Decree 199/2001) [7]; (2) to develop a procedure for assessing the conservation status of the aforementioned habitat types according to European and Spanish guidelines for monitoring the Natura 2000 network; and (3) to map the flora of interest in Annexes II and IV of Directive 92/43/EEC, in the Royal Decrees of 2001 and 2011, and on the Red List of Spanish Vascular Flora. One novel feature of this cartographic study is the incorporation of the new syntaxa described prior to the implementation of the EU code, which enriches habitat diversity.

The five mapped sites of community interest are included almost entirely to the province of Ciudad Real. (1) Rivers of the middle Guadiana basin and slopes, code ES4220003, covers an area of 23,483.92 hectares and encompasses 19 municipalities. It has a high plant and animal species richness, particularly *Betula pendula* Roth subsp. *fontqueri* (Rothm) G. Moreno and Peinado. (2) Sierra the Canalizos, code ES4220013, covers an area of 24,564.21 hectares and is classified as for public–private use; it acts as a corridor between Montes de Toledo and Sierra Morena. (3) Sierras of Almadén-Chillón -Guadalmez, code ES4222015, has an area of 6612.07 hectares, and is designated as being primarily for public use. (4) Sierra Morena, code ES4220014, has an area of 134,206.27 hectares and is the largest of the SCI in the province of Ciudad Real. It is for public and private use, predominantly private, generally in the form of private game reserves. Its extensive vegetation cover and high rate of habitats and endemisms justifies its declaration as a SCI for incorporation in the Natura 2000 network. (5) Lagunas de Ruidera, code ES4210017, with 34,452.00 hectares, is a site of community interest in the provinces of Ciudad Real and Albacete, and comprises eight municipalities in the Campo de Montiel. This is a wetland of interest that extends among holm oak and juniper woodlands, the latter covered by special protection measures from the EU and the autonomous region of Castilla La Mancha.

## 2. Materials and Methods

Five SCIs (sites of community interest) with an area of 223,500 hectares in the provinces of Ciudad Real and Albacete in the central Iberian Peninsula were mapped and studied using the phytosociological approach: (A) Almadén-Chillón y Guadalmez, (B) Sierra de los Canalizos, (C) Sierra Morena, (D) Guadiana y laderas vertientes, and (E) Lagunas de Ruidera. The vegetation was interpreted using phytosociological inventories, taken following the Braun–Blanquet phytosociological methodology [8], where the abundance scale combines an estimate of the number of individuals from each species and the area they occupy in the relevé area. The quantitative indexes and their values are as follows: + (from 0.1% to 1%); 1 (from 1% to 10%); 2 (from 10% to 25%); 3 (from 25% to 50%); 4 (from 50% to 75%); and 5 (from 75% to 100%), which we obtained from works on nearby territories [9–19]. The inventories were included in the corresponding habitat following the "Interpretation Manual of European Union Habitats (2013)" with the four-digit code, and the six-digit code of Habitats of Spain. The syntaxa described after the implementation of the six-digit code are shown as xxxxxx. The syntaxa of interest published after the introduction of the six-digit code in Spain, although they have been accepted by the scientific community, still have no code. They are therefore only known by the four-digit code of the Habitats Directive and a code xxxxxx, indicating this absence. The vegetation units were obtained from aerial photographs at a scale of 1:10,000, digital terrain models (which were used to generate an altitude, an orientation and a slope model), bioclimatic and biogeographical models, and geological and soil maps, which were subsequently verified in the field by taking phytosociological inventories of the plant associations present in each vegetation unit. All the databases were implemented in a geographic information system (GIS) [20–22].

## 3. Results

From the habitat maps made in the five SCIs, we obtained a range of habitats dominated by species in the *Quercus* genus: 9230—Mediterranean-Ibero-Atlantic and Galaico-Portuguese oak woods of *Quercus robur* and *Quercus pyrenaica* [23] (Figure 1a–c); 9240—*Quercus faginea* and *Quercus canariensis* Iberian woods (Figure 2a,b); 9320—Thermomediterranean forests of *Olea* and *Ceratonia* (Iberian Peninsula, Balearic and Canary Islands) (Figure 3); (Habitat of the Atlas and Manual of the Habitat of Spain: 832010 *Querco rotundifoliae-Oleion sylvestris* Barbero, Quézel, and Rivas-Martínez in Rivas-Martínez, Costa and Izco 1986. xxxxxx *Asparago albi-Oleetum sylvestris* Cantó, Ladero, Pérez Chiscano and Rivas-Martínez in Rivas-Martínez et al. 2011; 9330 - *Quercus suber* forests (Figure 4a–c); 9340—*Quercus ilex* and *Quercus rotundifolia* forests (Figure 5); 9561—Endemic forests of *Juniperus* spp. (* = priority habitat by the European Union) (Figure 6); 9540 - Mediterranean pine forests of endemic *Pinus pinaster* (Figure 7); 5210—Arborescent scrubs of *Juniperus* spp. (Figure 8a–c).

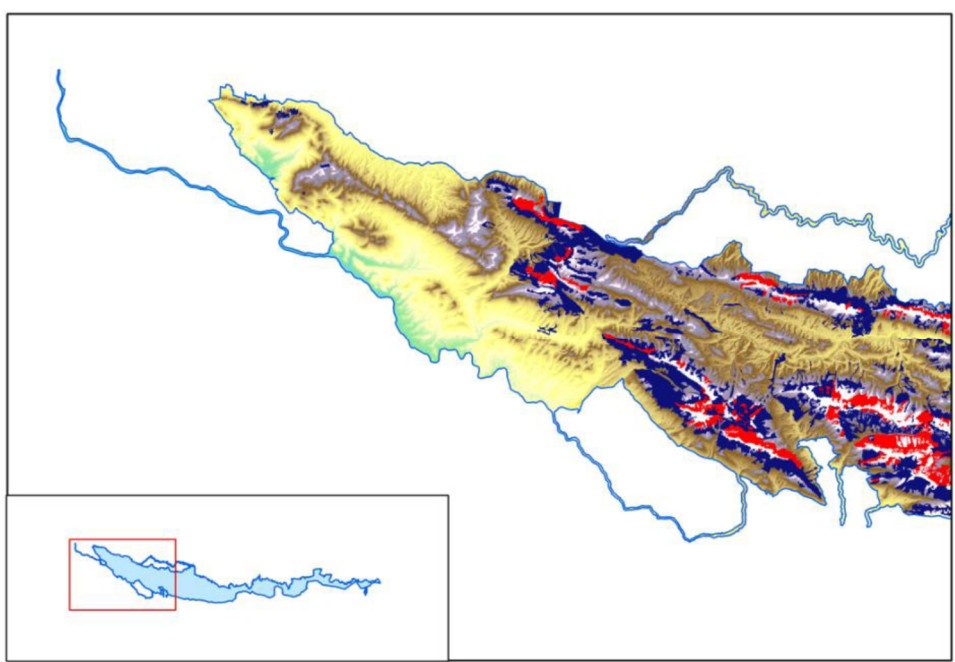

(**a**) SCI: Sierra Morena. Mapped associations (habitat). Blue: 9230, 823022. Red: 9230, 823029

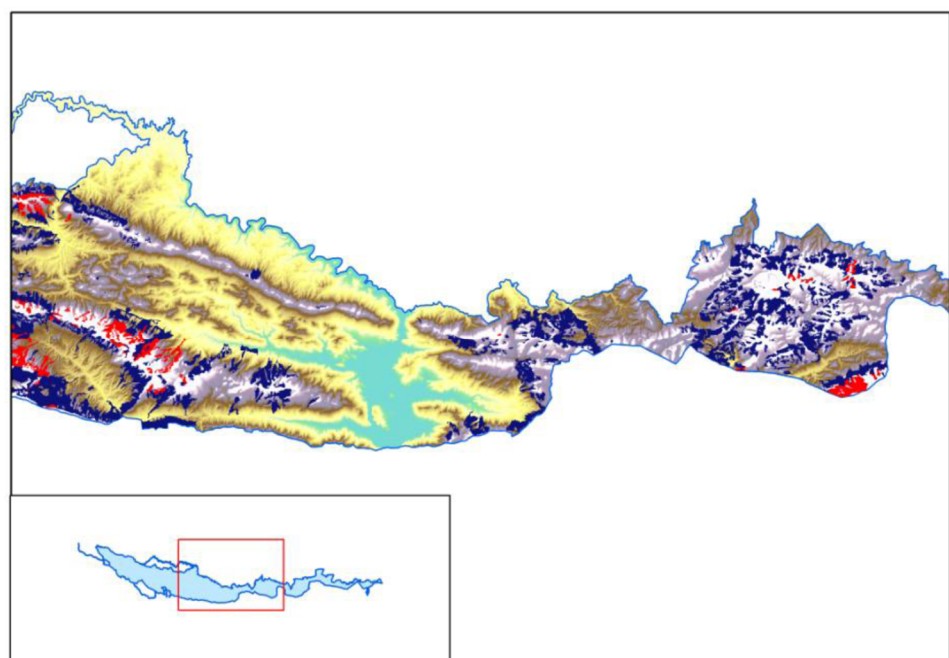

(**b**) SCI: Sierra Morena. Mapped associations (habitat). Blue: 9230, 823022. Red: 9230, 823029

**Figure 1.** *Cont.*

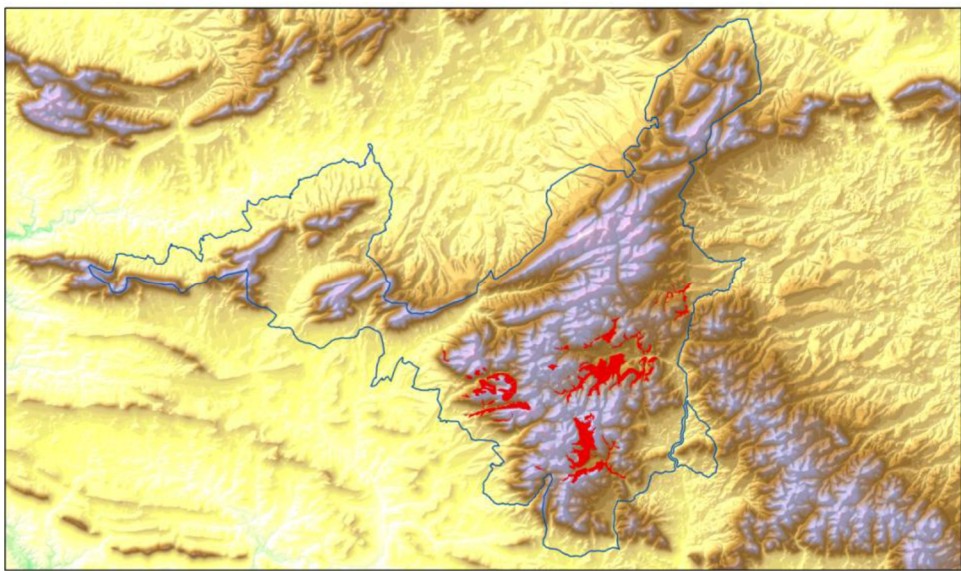

(**c**) SCI: Sierra the Canalizos. Mapped associations (habitat). Red: 9230, 823022

**Figure 1.** (**a–c**) 9230 Oak woods with *Quercus pyrenaica* and oak woods with *Quercus robur* and *Quercus pyrenaica* in the north-west Iberian Peninsula. Longitude 4°36′37″ W, latitude 38°56′38″ N (Saceruela). 823022 *Arbuto unedonis-Quercetum pyrenaicae* (Rivas Goday in Rivas Goday, Esteve, Galiano, Rigual & Rivas-Martínez 1960) Rivas-Martínez 1987. (Sierra de Canalizos, red polygons. Sierra Morena, blue polygons). 823029 *Sorbo torminalis-Quercetum pyrenaicae* Rivas Goday ex Rivas-Martínez 1987. (Sierra Morena, red polygons). The map shows deciduous forests of *Quercus pyrenaica* in two plant associations in sites of community interest.

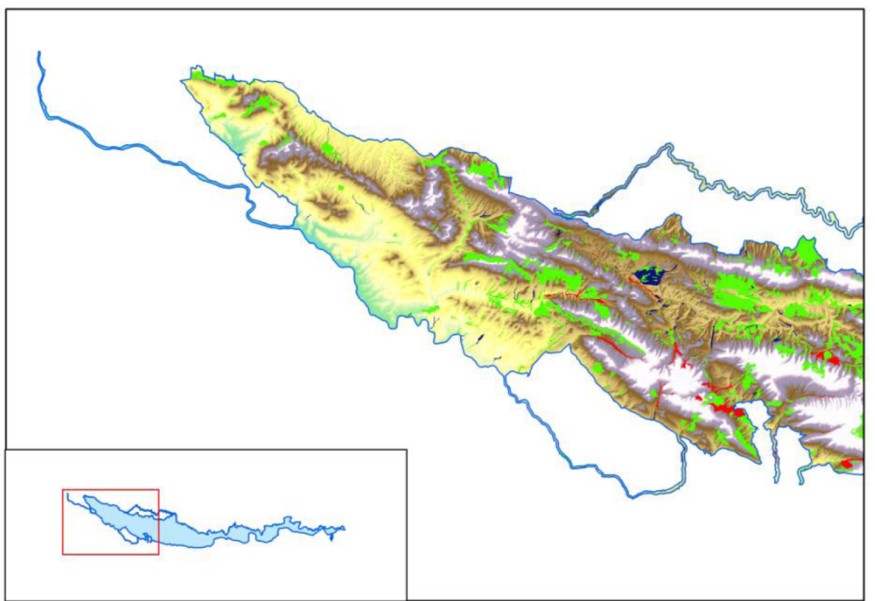

(**a**) SCI: Sierra Morena. Mapped associations (habitat). Blue: 9240, 824030. Red: 9240. Green: 9240, 824031

**Figure 2.** *Cont.*

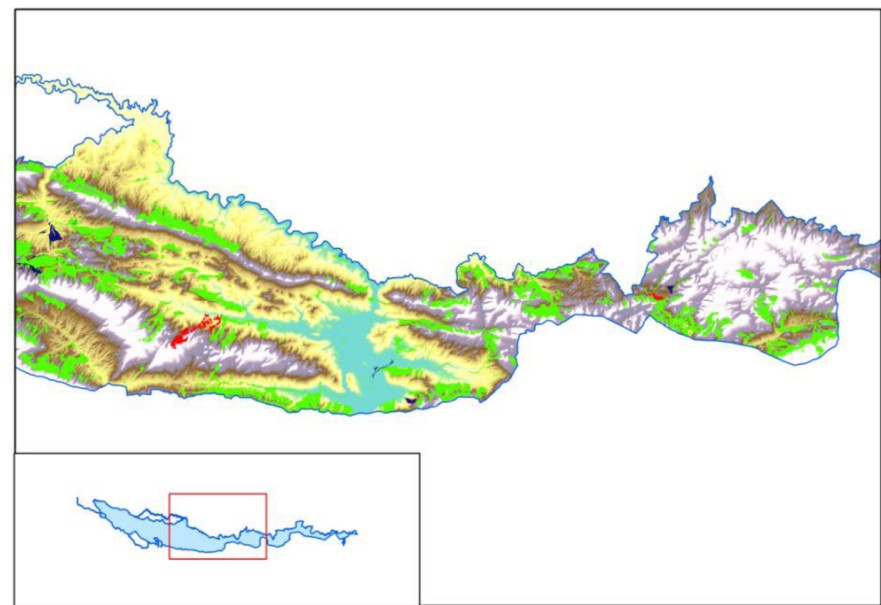

(**b**) SCI: Sierra Morena. Mapped associations (habitat). Blue: 9240, 824030. Red: 9240. Green: 9240, 824031

**Figure 2.** (**a**,**b**) 9240 *Quercus faginea* and *Quercus canariensis* Iberian woods. Longitude 4°18′16″ W, latitude 38°24′24″ N (Fuencaliente). 824030 *Quercion broteroi* Br.-Bl., P. Silva and Rozeira, 1956 em. Rivas-Martinez, 1975 corr. Ladero 1974. (Sierra Morena, blue polygons). 824031 *Pistacio terebinthi-Quercetum broteroi* Rivas Goday in Rivas Goday, Borja, Esteve, Galiano, Rigual, and Rivas-Martínez 1960. (Sierra Morena, green polygons). Xxxxxx *Pyro bourgaeanae-Quercetum broteroi* Cano, García-Fuentes, Torres, Pinto-Gomes, Cano-Ortiz, Montilla, Muñoz, Ruiz, and Rodríguez 2004. Xxxxxx *Doronico plantaginei-Quercetum canariensis* Rivas-Martínez and Cano 2011. The map shows the marcescent forests of Portuguese oak, *Quercus faginea* and *Quercus canariensis* in three plant associations in sites of community interest.

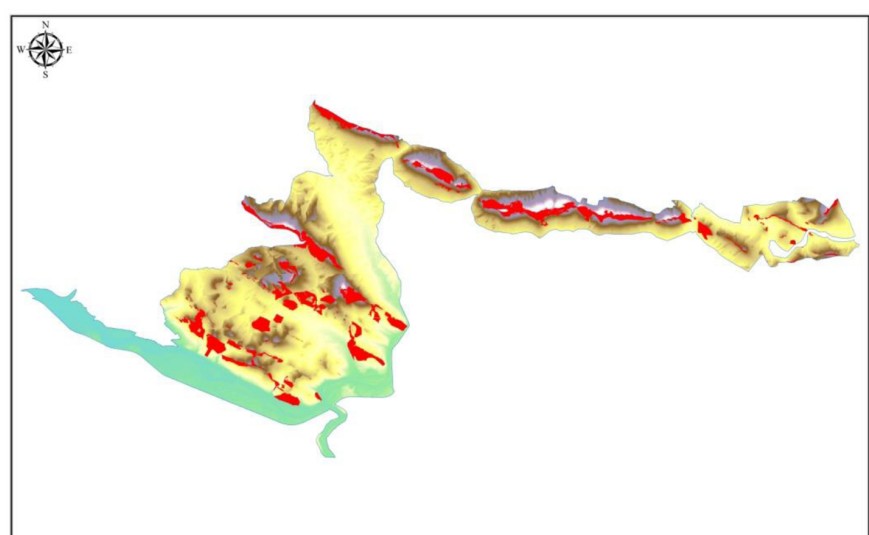

**Figure 3.** 9320 Thermomediterranean forests of *Olea* and *Ceratonia* (Iberian Peninsula, Balearic and Canary Islands). 832010 *Querco rotundifoliae-Oleion sylvestris* Barbero, Quézel and Rivas-Martínez in Rivas-Martínez, Costa, and Izco 1986. (Sierras de Almaden, Chillon and Guadalmez, red polygons). Xxxxxx *Asparago albi-Oleetum sylvestris* Cantó, Ladero, Pérez Chiscano, and Rivas-Martínez in Rivas-Martínez et al. 2011. The map shows the thermophilous forests of wild olive, *Olea europea* subsp. *sylvestris* (Mill.) Hegi, in a single plant association in sites of community interest. Longitude 4°50′24″ W, latitude 38°46′31″ N (Almaden).

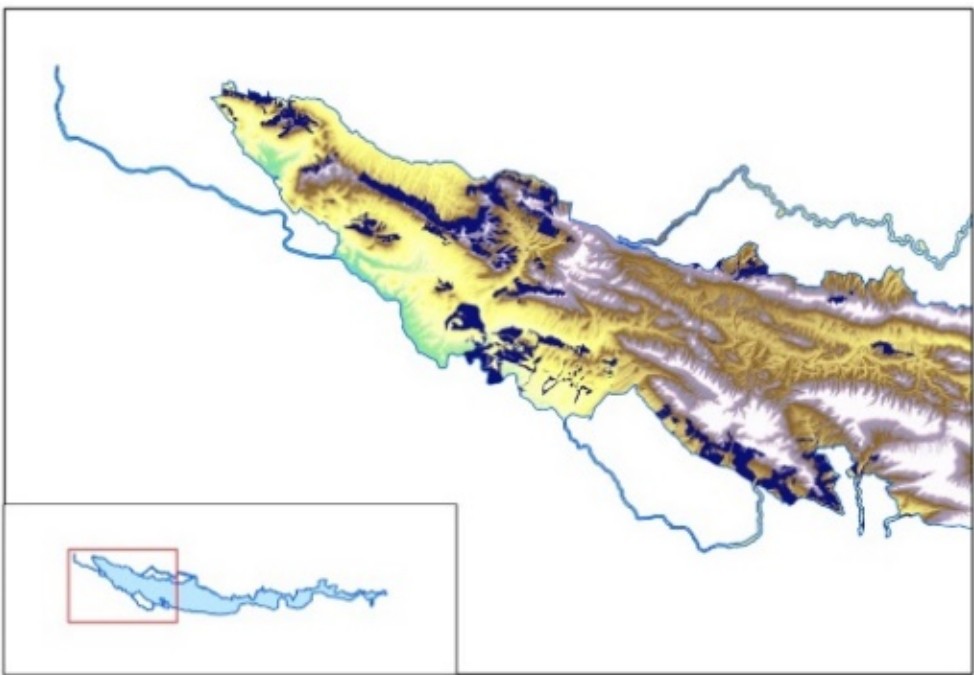

(**a**) SCI: Sierra Morena. Mapped associations (habitat). Blue: 9330, 833013

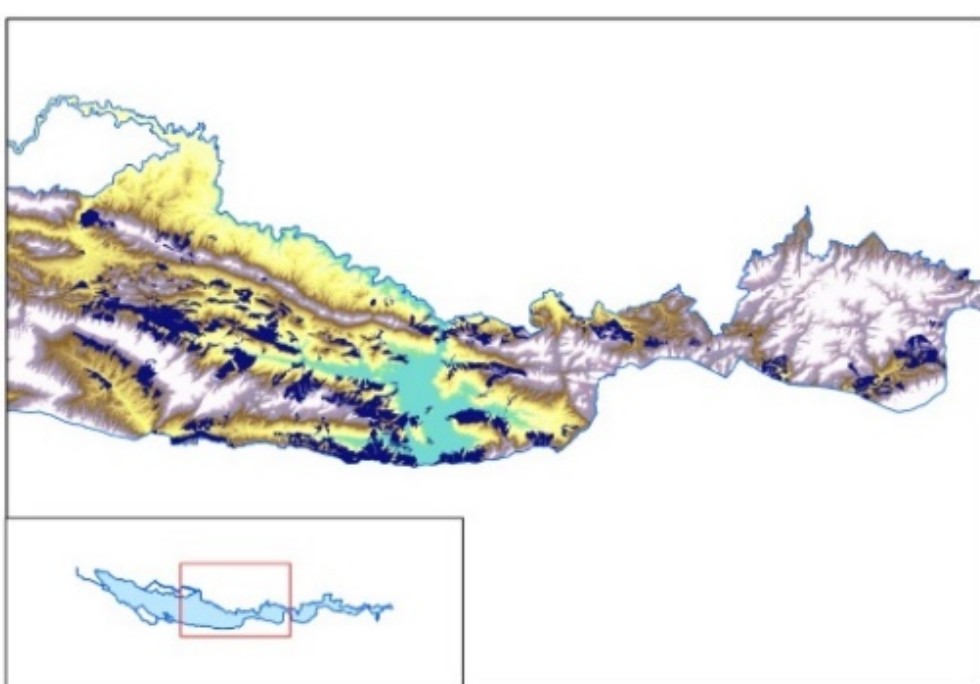

(**b**) SCI: Sierra Morena. Mapped associations (habitat). Blue: 9330, 833013

**Figure 4.** *Cont.*

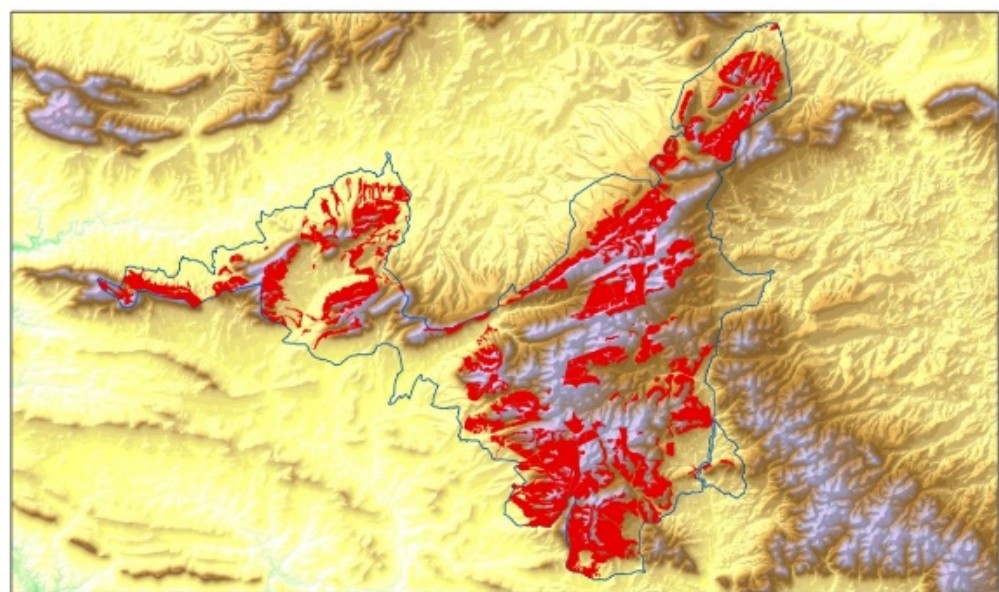

(**c**) SCI: Sierra the Canalizos. Mapped associations (habitat). Red: 9330, 833013

**Figure 4.** (**a**–**c**) 9330 *Quercus suber* L. woods (cork oak woods). Longitude 4°36′37″ W, latitude 38°56′38″ N (Saceruela). 833013 *Poterio agrimonioidis-Quercetum suberis* Rivas Goday in Rivas Goday, Borja, Esteve, Galiano, Rigual, and Rivas-Martínez 1960. (Sierra Morena, blue polygons; Sierra de Canalizos, red polygons). The map shows the forests of cork oak, *Quercus suber*, in a single plant association in sites of community interest. These forests are always located in the mesomediterranean thermotype.

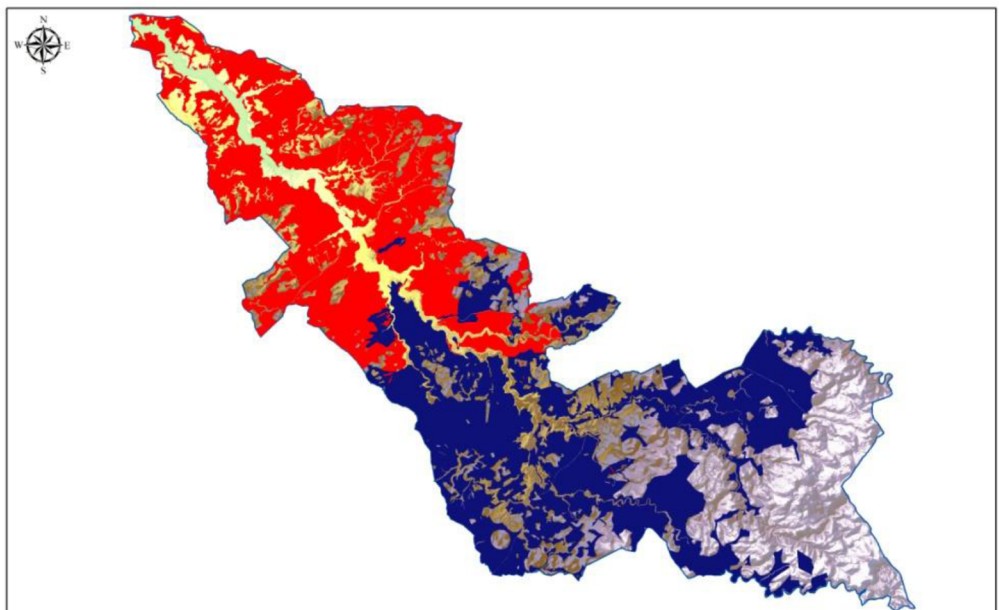

**Figure 5.** 9340 *Quercus ilex* L. and *Quercus rotundifolia* woods. Longitude 2°54′34″ W, latitude 38°57′34″ N (Lagunas de Ruidera). 834034 *Asparago acutifolii-Quercetum rotundifoliae* Rivas-Martínez, Cantó, Fernández-González, and Sánchez Mata in Rivas-Martínez et al. 2002. (Lagunas de Ruidera, red polygons). 834033 *Junipero thuriferae-Quercetum rotundifoliae* Rivas-Martínez 1987. (Lagunas de Ruidera, blue polygons). Included are the maps made in the different SCIs, silicicolous, and basophilous holm oak forests in both the mesomediterranean and the supramediterranean, with these latter acting as a transition to the juniper forests of *Juniperus thurifera* L.

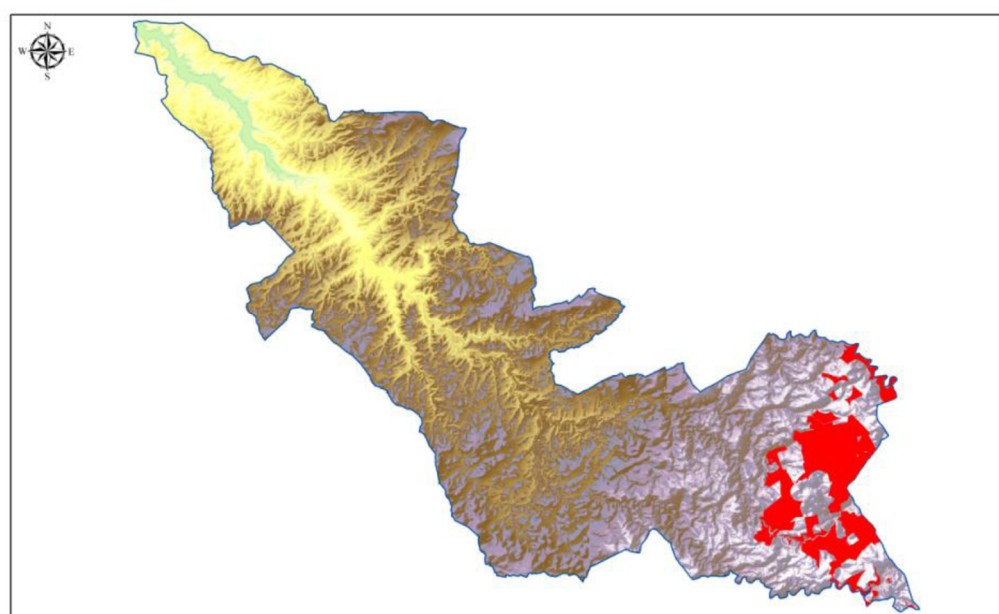

**Figure 6.** 9561 Endemic forests of *Juniperus* spp. (\* = priority habitat). 856112 *Juniperetum phoeniceo-thuriferae*. (Lagunas de Ruidera, red polygons). These juniper forests of *Juniperus thurifera* have a high ecological and botanical value and are protected in Spain by both European and Spanish national and regional regulations. Longitude 2°54′34″ W, latitude 38°57′34″ N (Lagunas de Ruidera).

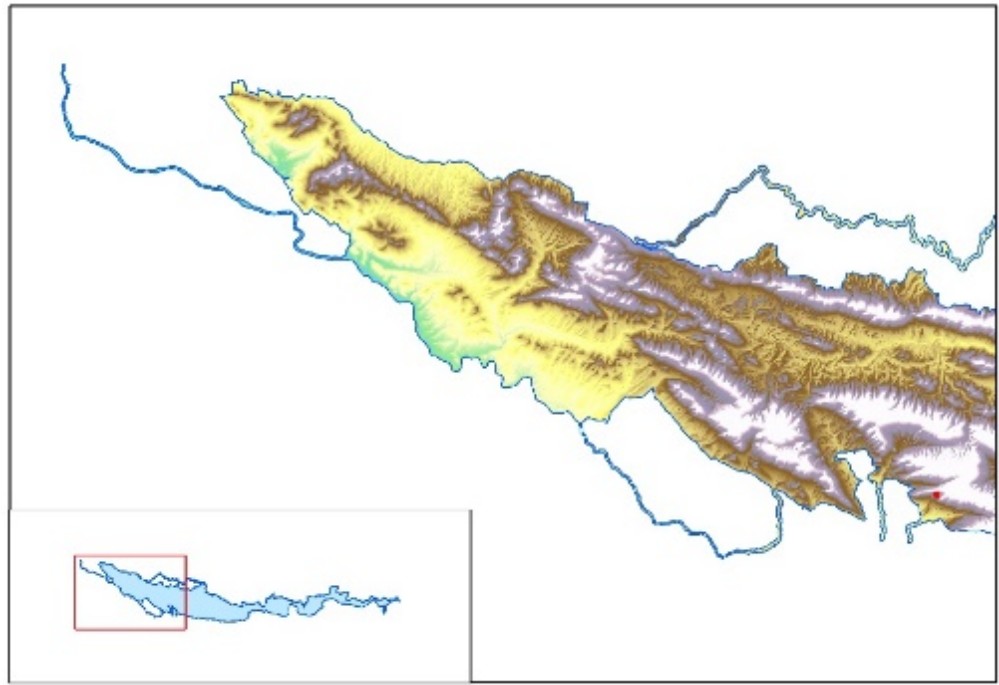

**Figure 7.** 9540 Mediterranean pine forests of endemic *Pinus pinaster* (*Pinus pinaster* subsp. *acutisquama*). (Sierra Morena, red polygons). In this case the map shows a pine forest community, of interest due to its presence in siliceous territories in the Sierra Morena. Longitude 4°18′16″ W, latitude 38°24′24″ N (Fuencaliente).

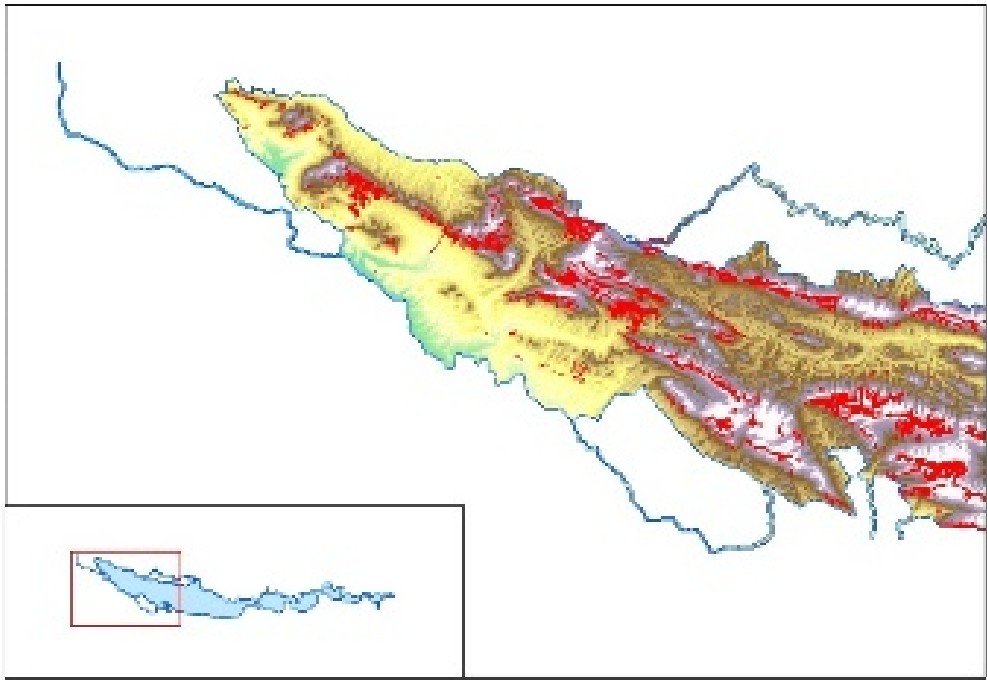

(**a**) SCI: Sierra Morena. Mapped associations (habitat). Red: 5210

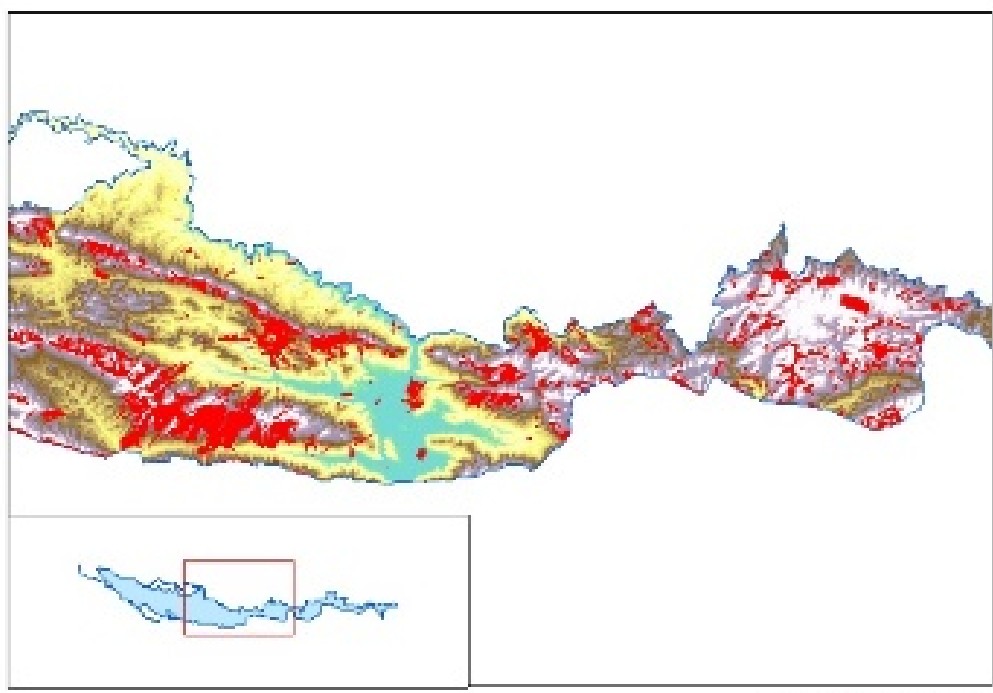

(**b**) SCI: Sierra Morena. Mapped associations (habitat). Red: 5210

**Figure 8.** *Cont.*

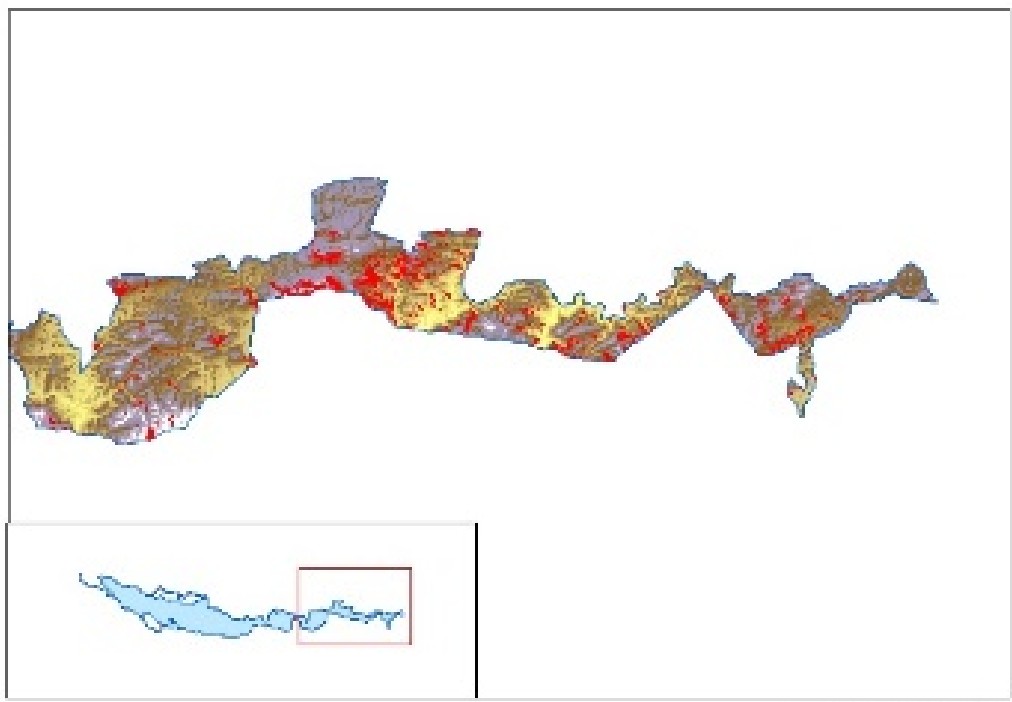

(**c**) SCI: Sierra Morena. Mapped associations (habitat). Red: 5210

**Figure 8.** (**a–c**) 5210 Arborescent scrubs of *Juniperus* spp. (*Juniperion badiae*). (Sierra Morena, red polygons). Communities of *Juniperus* spp. (*oxycedrus* group) are frequent on the Iberian Peninsula, in this case located in small patches on rocks. The patches that have been mapped are of great interest as they contain a high number of the endemic species found in this habitat. Longitude 4°4′16″ W, latitude 38°34′28″ N (Mestanza).

## 4. Discussion

*4.1. 9230: Mediterranean-Ibero-Atlantic and Galaico-Portuguese Oak Woods of* Quercus robur *and* Quercus pyrenaica

This habitat is frequent in shaded areas of the Sierra Morena and includes Pyrenean oak woods (*Quercus pyrenaica*). These are marcescent woods growing on deep and generally well-drained siliceous soils in subhumid-humid ombrotypes. The habitat is represented by mesomediterranean (*Arbuto unedonis-Quercetum pyrenaicae*) and supramediterranean (*Sorbo torminalis-Quercetum pyrenaicae*) Pyrenean oak woods. The association *Arbuto unedonis–Quercetum pyrenaicae* corresponds to a multi-layered and marcescent woodland, with a similar physiognomy to a Pyrenean oak woodland, and which grows on moderately acid soils that are rich in organic matter of plant origin, provided there is sufficient moisture to offset or minimize the period of summer drought. It therefore tends to grow on very strong soils with a high water retention capacity and in favorable topographic situations such as slopes with little or no sun, valley floors, at the foot of escarpments, and others. This luso-extremaduran woodland is characteristic of the subhumid–humid mesomediterranean belt.

The *Sorbo torminalis–Quercetum pyrenaicae* is a wood in which as many as five layers can be distinguished: tree, shrub, vine, herbaceous, and moss. The tree layer has very high cover; the shrub layer is not very dense; and the herbaceous layer has abundant early-flowering perennial bulbs and plants with a marked ombrophilous character. It thrives on humic cambisols and chromic luvisols originating from slate and quartzite, preferentially in shaded orientations. Its presence is linked not only to the availability of high humidity, but also to a shorter period of summer drought. This community is characteristic of the Mediterranean oceanic pluviseasonal bioclimate in the supra-Mediterranean subhumid–humid bioclimatic belt [24].

*Sorbo torminalis–Quercetum pyrenaicae* (Figure 9) represents the mature stage of the luso-extremaduran supramediterranean neutral–acidophilous subhumid–humid climatophilous

series of *Q. pyrenaica* woods with *Sorbus torminalis* (L.) Crantz (*Sorbo torminalis–Querco pyrenaicae* S.) The shrub formations that substitute the Pyrenean oak wood with whitebeam comprise stands of *Genista florida* L. (*Genisto floridae–Adenocarpetum argyrophylli*) and—due to the greater soil degradation and loss—heaths or rock rose-heaths of *Erica umbellata* and *E. australis* (*Halimio ocymoidis–Ericetum umbellatae)*, and broom stands of *Cytisus scoparius* subsp. *bourgaei* (Boiss.) R. Mart., Fern. Gonz., and Sánchez Mata (*Adenocarpo telonensis–Cytisetum bourgaei*). Worth noting for their abundance among the serial herbaceous communities are the marianense grasslands (*Avenulo occidentalis–Festucetum elegantis*), whose faithful species (*Festuca elegans* Boiss.) is of community interest [25].

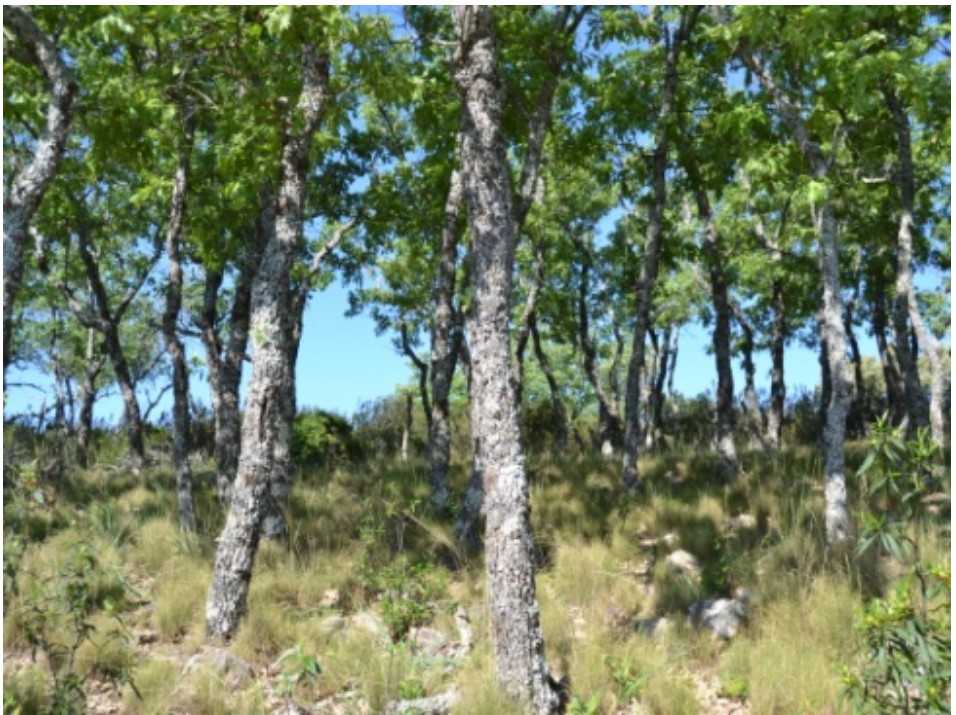

**Figure 9.** Supramediterranean Pyrenean oak wood of *Quercus pirenaica* in Sierra Morena. *Sorbo torminalis–Quercetum pyrenaicae* with grassland of *Avenulo occidentalis–Festucetum elegantis*.

The mesomediterranean Pyrenean oak wood constitutes the head of the lusoextremaduran mesomediterranean acidophilous and neutral–acidophilous subhumid–humid climatophilous series of *Q. pyrenaica* woodlands with *Arbutus unedo* (*Arbuto unedonis–Querco pyrenaicae*). The substitution stages are also slightly modified. There is an incorporation of formations of *Arbutus unedo* (*Phillyreo angustifoliae–Arbutetum unedonis*) (Figure 10), the heaths, rock rose-heaths, and broom formations of *C. scoparius* subsp. *bourgaei* are maintained, and the shrub formations of *G. florida* and the marianense grasslands of *Avenulo occidentalis–Festucetum elegantis* disappear. The understorey is also somewhat impoverished in nemoral herbaceous elements compared to its supramediterranean equivalents, but this is counteracted by the participation of other typical sclerophyllous woodland species. In the understorey of cleared or juvenile formations and in boundary areas, there is often a presence of elements of *Origanion virentis*.

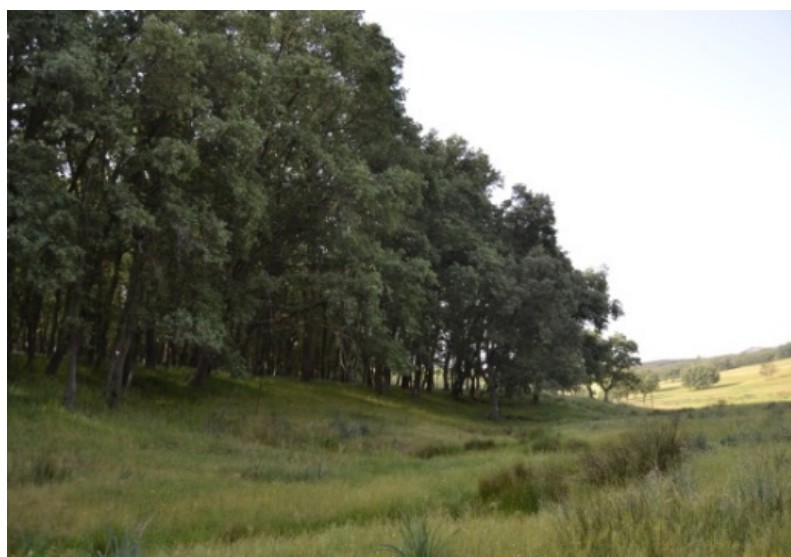

**Figure 10.** Mesomediterranean Pyrenean oak wood of *Quercus pyrenaica* in Sierra Morena (El Viso). *Arbuto unedonis–Quercetum pyrenaicae*.

Habitat Conservation Status

The Pyrenean oak wood with *Arbutus unedo* currently presents well-conserved stands in the eastern Sierra Morena and in some other SCIs; however, most have been replaced by pine reforestations in the more accessible zones. The Pyrenean oak woods with whitebeam are located preferentially in shady depressions and stony areas in the main mountain ranges above altitudes of 1000 m, although nowadays they are frequently found in a mosaic with characteristic communities of the series. The *Quercus pyrenaica* woods in the Sierra Morena are generally in a good state of conservation, although in certain places they have been subjected to deforestation, leading to the installation of secondary woodland. These woods must be conserved to avoid the destruction of the palaeosoils, which otherwise tend to disappear. The association *Arbuto unedonis–Quercetum pyrenaicae* has a predominance of *Q. pyrenaica* in the tree layer, usually accompanied by species such as *Q. faginea* subsp. *faginea* and *Q. faginea* subsp. *broteroi*, in addition to *Acer monspessulanum* L. Other characteristic species of the community include *Arbutus unedo, Crataegus monogyna* Jacq., and *Tamus communis* L. among others. The following companion species are also present: *Erica arborea* L., *Viburnum tinus* L., *Pistacia terebinthus* L., and *Phillyrea angustifolia* L., etc.

The association *Sorbo torminalis–Quercetum pyrenaicae* is characterized, among other species, by *Q. pyrenaica, Sorbus torminalis, S. aria, Acer monspessulanum, Allium massaessylum* Batt. and Trab., and *Milium montianum* Parl. Companion species include *E. arborea, Cytisus scoparius* (L.) Link, *Rubus ulmifolius* Schott, *Rosa canina* L., etc. It is also worth noting the presence in some Pyrenean oak woods of *Arbuto unedonis–Quercetum pyrenaicae* of *Prunus lusitanica* L. and *Sorbus torminalis* (L.) Crantz; and of *Narcissus bulbocodium* L. in *Sorbo torminalis-Quercetum pyrenaicae*. The Pyrenean oak wood of *Sorbo torminalis–Quercetum pyrenaicae* has a relict character and constitutes islands of Atlantic influence in a typically Mediterranean environment. It has significant ecological value, as it serves as a refuge for certain plant and animal species and communities, and is found in steeply sloping siliceous mountain ranges with soils that are prone to erosion.

*4.2. 9240:* Quercus faginea *and* Quercus canariensis *Iberian Woods*

This habitat (9240) accompanies the Mediterranean sclerophyllous woods dominated by holm oak (*Quercus rotundifolia* Lam. = *Q. ilex* subsp. *ballota* (Desf.) Samp.), in a continental and more or less dry climate, or by *Quercus ilex* L. subsp. *ilex*, in an oceanic and more humid climate (Habitat 9340). In the SCI, it is frequently located in foothills, riverbeds, and very humid areas, generally representing the transitional stage between cork oak and common

oak woods. This habitat contains the luso-extremaduran Portuguese oak woods with *Pistacia terebinthus* and *Arbutus unedo*, and *Pyro bourgaeanae–Quercetum broteroi*, (Figures 11 and 12), which is present in more easterly territories and therefore with less Atlantic influence, on deep siliceous soils in a subhumid ombrotype with a continental influence, and on soils subject to temporary waterlogging. The same territories contain the marianense mesomediterranean subhumid–humid pluviseasonal silicicolous climato-temporihygrophilous series of *Quercus canariensis* and *Doronicum plantagineum* woods [26–30].

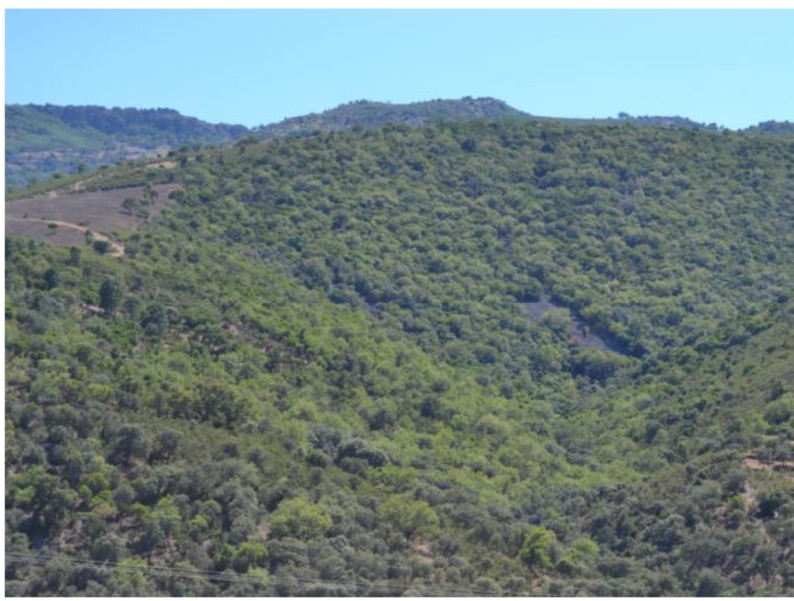

**Figure 11.** *Pistacio terebinthi–Quercetum broteroi*.

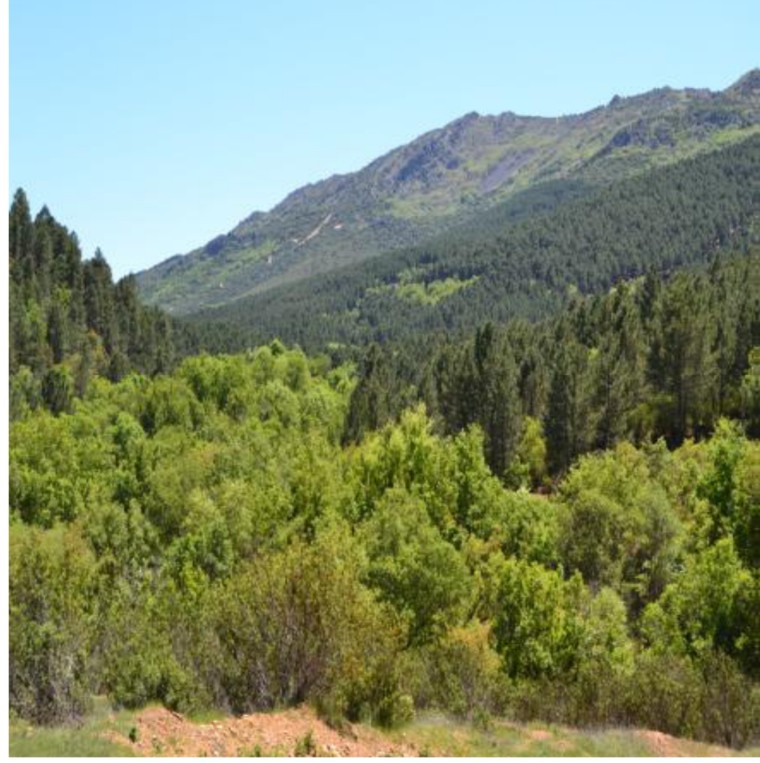

**Figure 12.** *Pyro bourgaeanae–Quercetum broteroi*.

### 4.3. 9330: Quercus suber *Forests*

This habitat is frequently found in shaded areas in several SCIs, becoming less common in the easternmost areas of the SCI due to declining rainfall and increasing continentality. The habitat is characterized by sclerophyllous oak whose tree layer is dominated by the cork oak (*Quercus suber*), but with the incursion of other tree species (*Q. rotundifolia, Q. broteroi*) or wild olive (*Olea sylvestris*) [31].

The understory of the cork oak woods is rich in shrubby plants such as *Arbutus unedo*, *Phillyrea angustifolia*, *Pyrus bourgaeana* Decne, *Viburnum tinus* L., *Myrtus communis* L., *Pistacia lentiscus* L., and *Phillyrea latifolia* L. There is also a frequency of climbing plants, particularly those belonging to the genera *Lonicera, Asparagus, Tamus, Hedera*, and *Smilax*.

They are normally found on deep cool soils originating from acid rocks (granite, schists, etc.), and in fairly warm (meso-Mediterranean) and humid (subhumid–humid) territories, where they represent the climactic forest. These SCIs contain particularly the following typical and characteristic species: *Q. suber, Arisarum vulgare* Targ.Tozz., *Asparagus acutifolius* L., *Daphne gnidium* L., *Rubia peregrina* L., *Smilax aspera* L., *Vincetoxicum nigrum* (L.) Moench, *Asplenium onopteris* L., *Phillyrea latifolia, Rosa sempervirens* L., *Ruscus aculeatus* L., *V. tinus, Arum italicum* Mill., *Sanguisorba hybrida* (L.) Font Quer, *Pyrus bourgaeana*, and *Lonicera* spp. The habitat is considered to have a highly altered floristic structure in these SCIs (Figures 13 and 14).

Habitat Conservation Status

In their optimal state, these are dense and closed woodlands that may reach a height of 15–25 m. Cork is one of the most economically valuable products derived from these woodlands. Other products with a potential economic return include livestock, wild mushrooms, and game. From the ecological point of view, they play an important role in $CO_2$ retention, regulating the water cycle, producing and retaining the soil, and regularizing the local climate. This habitat is home to a large number of typical Mediterranean woodland species, including several floristic species of importance for conservation. To this must be added its scenic, scientific, and educational value. The SCI contains only a few small remnants of this woodland, highly structurally altered, and with clearly compromised socio-economic functions. Very few examples survive today, and those that do are fragmented and with a far from optimal floristic composition. It is considered globally to have an average conservation status.

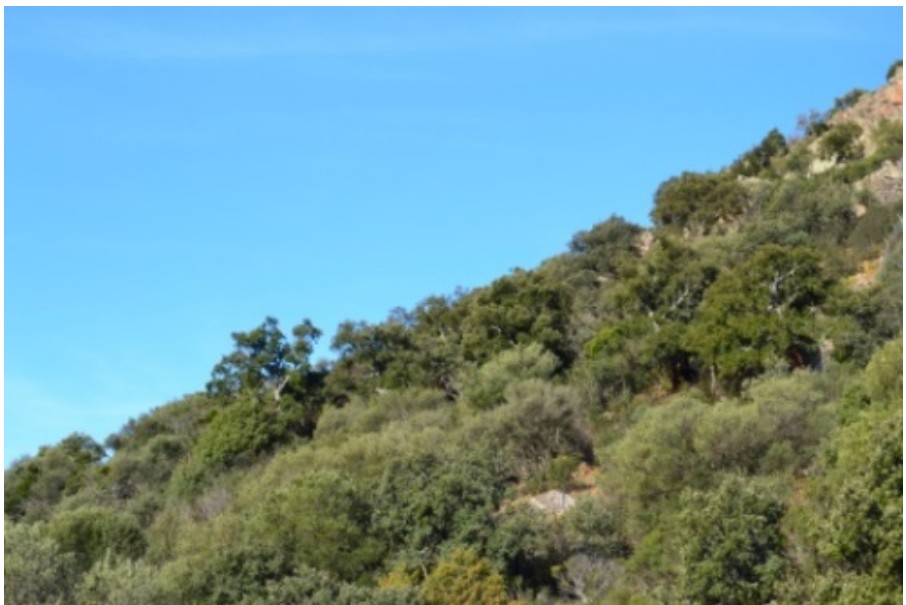

**Figure 13.** *Poterio agrimonioidis–Quercetum suberis.*

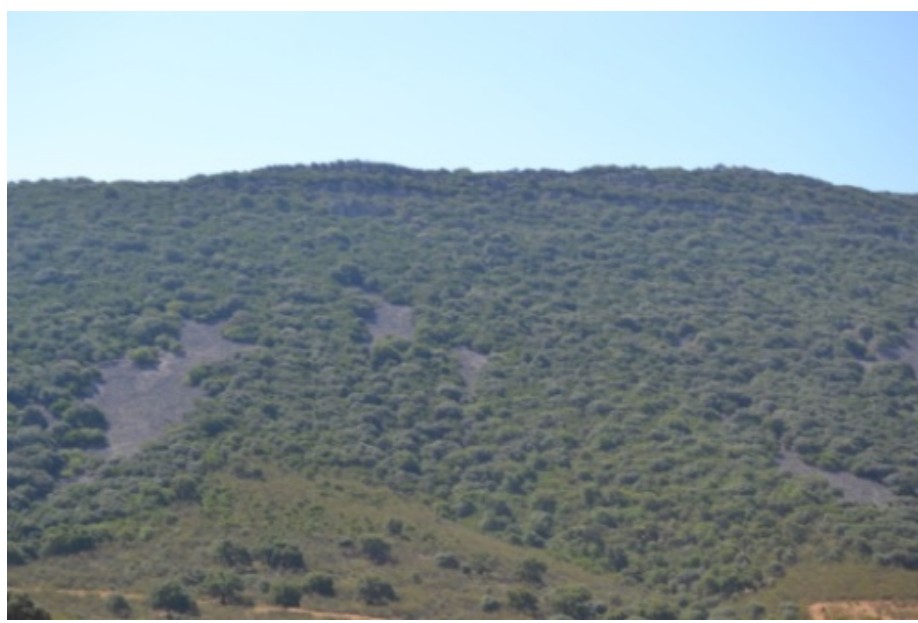

**Figure 14.** *Poterio agrimonioidis–Quercetum suberis.*

*4.4. 9340:* Quercus ilex *and* Quercus rotundifolia *Woods*

This habitat includes the holm oak woods in the five SCIs, four of which are on acid substrates and one on basic substrates. In the case of acid substrates, these are dense sclerophyllous woods with a tree layer dominated by the Castilian holm oak [*Quercus rotundifolia = Q. ilex* subsp. *ballota* (Desf.) Samp.], where other trees such as *Olea sylvestris, Pyrus bourgeana,* and others may still be present. As with other Mediterranean woodlands, they are rich in vines. The most common are *Smilax aspera, Bryonia dioica* Jacq., *Tamus communis, Rubia peregrina, Vincetoxicum nigrum, Aristolochia pistolochia* L., *Hedera* spp., *Lonicera periclymenum* L., and *Clematis flammula* L. The shrub layer frequently contains sclerophyllous species, some broadleaved, such as *Viburnum tinus, Phyllirea latifolia, Ph. angustifolia, Pistacia terebinthus, Jasminum fruticans* L., *Rhamnus oleoides* L., *R. lycioides* subsp. *laderoi* Rivas Mart. and J.M. Pizarro, *Myrtus communis* and *Ruscus aculeatus*, among others. Holm oak woods can be found in almost all soil types with different depths, stoniness, texture types, pH, and nutrient availability. Their plasticity enables them to occupy both deep and skeletal soils, and they may even colonize rocky environments when the system of crevices allows the roots to access deep water reserves [32]. In terms of climate, these woodlands have a greater continental and xerophilous character than cork oaks. They are found in mesomediterranean and thermomediterranean territories, normally with low annual rainfall (dry to subhumid territories). From the dynamic point of view, they represent the climactic stage in a large part of the territory of these SCIs. The following typical and characteristic species are especially notable: *Q. rotundifolia, Q. suber, Asparagus acutifolius, O. sylvestris, Daphne gnidium, R. peregrina, S. aspera, V. nigrum, Ph. latifolia, Ph. angustifolia, R. aculeatus, V. tinus, J. fruticans, Sanguisorba hybrida, P. bourgaeana,* and *Lonicera* spp. (Figure 15).

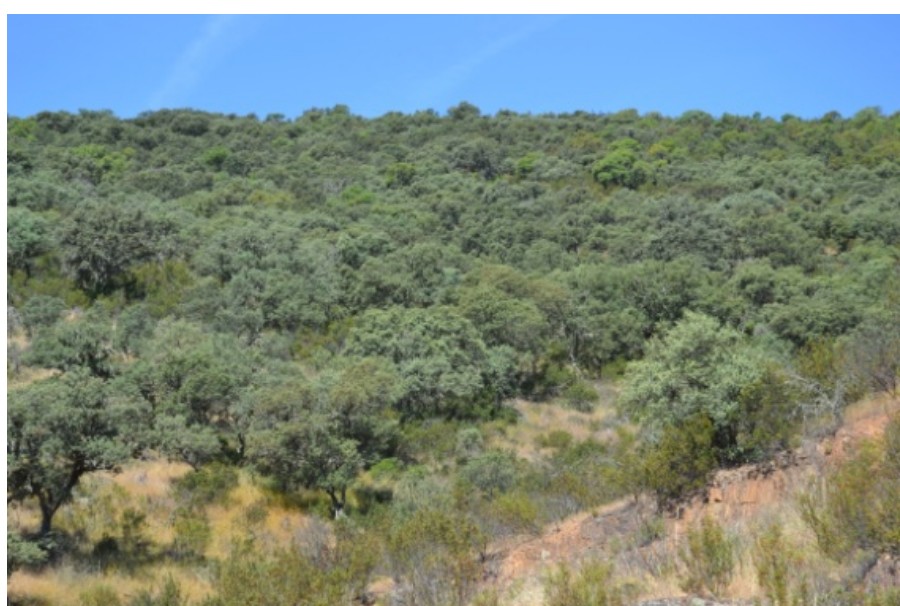

**Figure 15.** *Pyro bourgaeanae–Quercetum rotundifoliae.*

Habitat Conservation Status

In their optimal state, these are dense closed woodlands that may reach a height of 15 m. From the ecological point of view, they play an important role as a biodiversity refuge. Average- or high-density holm oak woods may be home to an abundance of animal species such as wild boar, roe deer, genet, marten, badger, fieldmouse, garden dormouse, sparrowhawk, goshawk, tawny owl, common wood pigeon, and jay, among others. More open holm oak woods may contain wolf, deer, fallow deer, Spanish imperial eagle, Bonelli's eagle, short-toed snake eagle, buzzard, European turtle dove, woodlark, woodchat shrike, Dartford warbler, Iberian magpie, the Montpellier snake, and more. There are also many insect species, including several associated to oak gall. Holm oak woods play an important role in retaining $CO_2$, regulating the water cycle, producing and retaining soil, and regularizing the local climate. To this must be added their scenic, scientific and educational value. The SCI contains barely a few small remnants of this woodland, highly structurally altered, and with clearly compromised socio-economic functions.

9320. Thermomediterranean forests of *Olea* and *Ceratonia* (Iberian Peninsula, Balearic and Canary Islands). 6310. Evergreen wooded pastures ("dehesa") with *Quercus*. Habitat 9320 includes the association *Asparago albi-Oleetum sylvestris* Cantó, Ladero, Pérez Chiscano, and Rivas-Martínez in Rivas-Martínez et al., 2011 (Figure 16), and is very well represented in the SCIs with thermophilous characteristics (Almadén-Chillón-Guadalmez and Sierra Morena) [27]. These are tall scrubland formations (almost 5 m in height) with varying degrees of openness. From the environmental standpoint, they are important for $CO_2$ fixation, soil protection, and preventing erosion, and for the sustenance of insectivorous birds during the winter months and the migration period. These communities of wild olive trees may be an excellent source of genetic diversity on which to develop new crop varieties. We highlight the following as typical and characteristic species: *Olea sylvestris, Asparagus albus* L., *A. acutifolius, Rhamnus lycioides* subsp. *laderoi, Arisarum simorrhinum* Durieu, *Pyrus bourgaeana, Daphne gnidium, Pistacia terebinthus, P. lentiscus, Retama sphaerocarpa* (L.) Boiss., *Cytisus bourgaei,* and *Lavandula stoechas* subsp. *sampaiana* Rozeira (Rey et al., 2009; Rivas-Martínez et al., 2011).

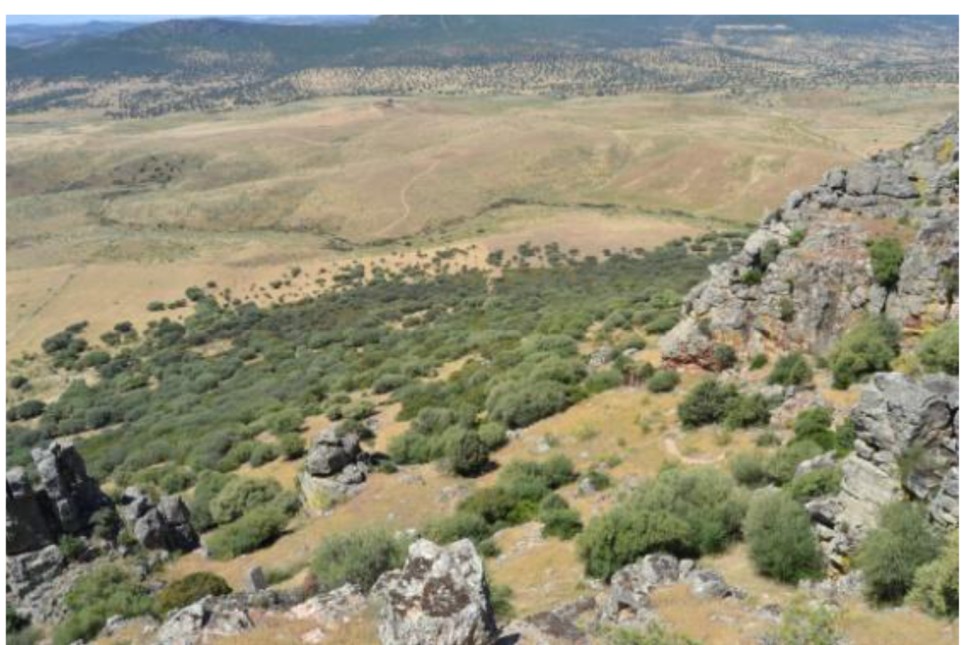

**Figure 16.** *Asparago albi-Oleetum sylvestris.*

### 4.5. Other Habitats of Interest

Habitat 6310 contains the evergreen wooded pastures ("dehesas") of great utility for livestock, including holm oak pastures (*Quercus rotundifolia*). "Dehesas" are seminatural habitats formed by perennial pastures and a sparse tree cover, the result of clearing old forests (in this case holm oak forests) [33]. This habitat is anthropic in origin, characterized by and dependent on extensive grazing by sheep, goats, and pigs, and may still contain plots with extensive rupestrian formations. The typical pastures are dominated by hemicryptophytes with a particular predominance of grasslands of *Poa bulbosa* L. included in Priority Habitat 6220*. Very sporadically they exist with other oak species, particularly *Q. faginea* subsp. *broteroi* and *Q. pyrenaica*.

Other habitats represent the catenal contact with the previous habitats, or else have been obtained through a dynamic with *Quercus* woods: 9540. Mediterranean pine forests with endemic Mesogean pines; 9560. Endemic forests with *Juniperus* spp.; 5210. Arborescent matorral with *Juniperus* spp.; 5330. Thermomediterranean and pre-desert scrub; 4030. European dry heaths.

Habitat 9540 is located exclusively in Sierra Morena, and consists of Mediterranean pine forests with endemic Mesogean pines. The maritime or cluster pine (*Pinus pinaster* Aiton) thrives on sandbanks and scree beds with varying degrees of acidity at altitudes of between 700 and 1700 m. In the SCI, it is located as a reduced population on the ridges of Navalmanzano, specifically at an altitude of 1035 m. This community is exclusive to the Sierra Morena SCI (Sierra Navalmanzano). At the floristic level, it is considered a monospecific population of *P. pinaster*, with infrequent representation of the genus *Juniperus*, but which we include in the alliance *Juniperion badiae* [34].

Habitat 9560, present in the Lagunas de Ruidera SCI, represents the juniper formations of *Juniperus thurifera* and *J. phoenicea* L., framed in the Iberian association of *Juniperetum phoeniceo-thuriferae* (Figure 17). These are arborescent microwoods usually with a very open tree and shrub layer, alternating with an herb layer of xerophilous grasslands. It is found in very dry environments, particularly in impoverished substrates and in climates where oak species cannot survive. The examples in this territory are differentiated from other Iberian juniper stands, particularly the taller ones, by the presence of thermophilous elements such as rosemary (*Rosmarinus officinalis* L.), Mediterranean buckthorn (*Rhamnus alaternus* L.), black hawthorn (*R. lycioides* L.), and jasmine (*Jasminum fruticans*) [35].

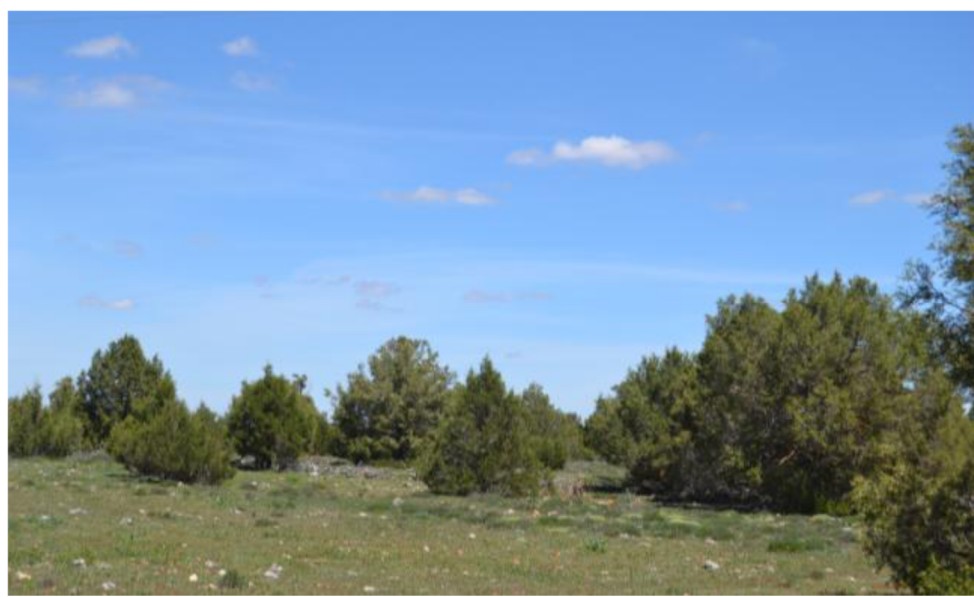

**Figure 17.** *Juniperetum phoeniceo-thuriferae.*

Other habitats of interest are the kermes oak stands and broom scrubland in 5330, whereas the juniper stands of *J. oxycedrus* subsp. *badia* (H. Gay) Debeaux are included in 5210 [36]. This habitat is characterized by tall scrublands, a Mediterranean and Submediterranean climate, and mid-mountain scrublands. This shrub belt is found above the forest levels or is present in clearings or degraded areas of forest, and characterized as cushiony mellipherous scrublands subjected to a degree of environmental dryness, dominated by low cushion-forming and frequently spiny scrublands. They have a variable relict character and are of great scenic value. Typical species in this habitat are *J. oxycedrus* subsp. *badia, J. oxycedrus* subsp. *oxycedrus, Echinospartum ibericum* Rivas Mart. Sánchez Mata and Sancho, *Q. rotundifolia, Phillyrea angustifolia, Arbutus unedo, Daphne gnidium, Thapsia villosa* L., and *Erica arborea* (Figure 18) [37,38].

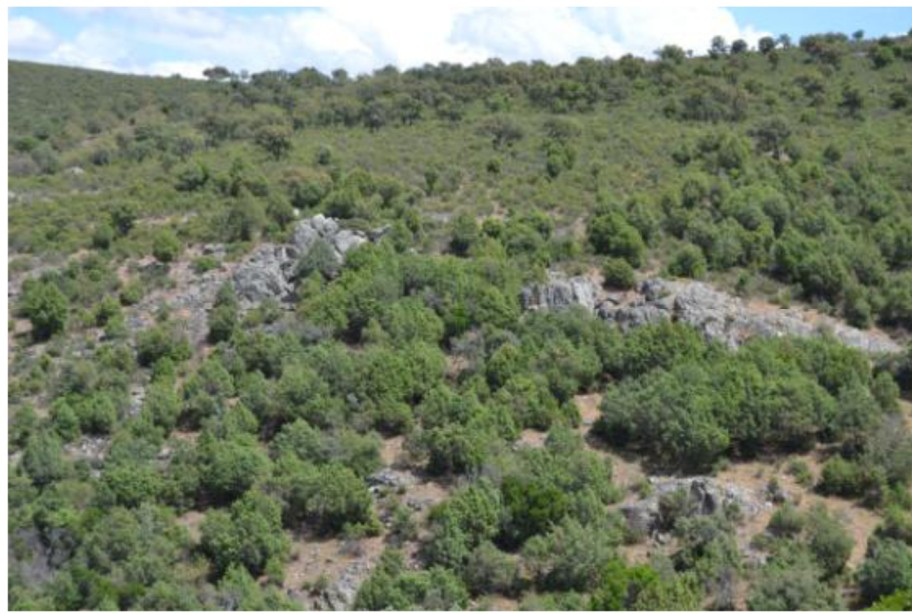

**Figure 18.** *Juniperion badiae.*

　　　　Finally, Habitat 4030 represents European dry heaths, and contains acidophilous scrub dominated by heaths and rock rose (heath and heath-rock rose formations). These are generally closed shrubby communities found in subhumid to humid climates with a marked Atlantic influence. Various heliophilous scrubs are dominated by *Erica australis*, *E. umbellata*, *E. scoparia* L., *Cistus populifolius* L., *Calluna vulgaris* (L.) Hull, *Halimium ocymoides* (Lam.) Willk., and *Pterospartum tridentatum* (L.) Willk, among others. Some of these scrublands are found on shallow oligotrophic soils that are degraded and acidified on the surface, as is the case of the nano-heaths of *E. umbellata* (community of *Halimio ocymoidis-Ericetum umbellatae*); they are found in the mesomediterranean subhumid-humid belt, where they constitute an advanced stage of degradation of cork oak and Pyrenean oak stands. In contrast, the rock rose-heath formations of *C. populifolius* (communities of *Erico australis-Cistetum populifolii* and *Polygalo microphyllae-Cistetum populifolii*) grow on somewhat structured and relatively well-conserved soils in foothills, and in temperate and cool shaded areas in upper thermomediterranean and mesomediterranean territories with a sub-oceanic influence and a subhumid-humid ombrotype. They generally represent a substitution stage of ombrophilous cork oak and Pyrenean oak formations. Finally, the heaths dominated by *E. scoparia* (community of *Lavandulo luisieri-Ericetum scopariae*) are found on sandy–silty soils subjected to occasional waterlogging in luso-extremaduran meso-supra-Mediterranean subhumid territories, and constitute an advanced stage of degradation of Pyrenean oak stands of *Q. faginea* subsp. *broteroi*. Typical species in this habitat are *E. australis* subsp. *australis*, *E. umbellata*, *E. scoparia*, *C. populifolius*, *Calluna vulgaris*, *Genista triacanthos* Brot., *Pterospartum tridentatum*, *Halimium ocymoides*, *Lavandula stoechas* subsp. *luisieri* (Rozeira) Rozeira, and *Polygala microphylla* L. [39,40]. The presence of *Drosophyllum lusitanicum* (L.) Link makes these heaths especially valuable for conservation. These scrublands have a high biomass and cover, with the exception of the nano-heaths of *E. umbellata*, which tend to have a height of around 50 cm, whereas the other species are medium to tall in height (80–170 cm).

　　　　This habitat has considerable floristic diversity. The global distribution of many of the dominant species is almost exclusively on the Iberian Peninsula (in some cases extending to southern France and/or northern Morocco), as is the case of *C. populifolius*, *E. umbellata*, *E. australis*, and *P. tridentatum*. Some of the species found here are Iberian endemisms such as *L. stoechas* subsp. *luisieri* and *Thymus mastichina* (L.) L. This habitat also plays a key role in honey production, as it contains several melliferous species. The detailed analysis of these habitats reveals a high diversity of woodlands rich in *Quercus* species, with differences and similarities in regard to the presence of habitats in the various SCIs (Table 1). We detected two groups of woodlands related to the genus *Quercus*; holm oak woods of *Q. rotundifolia*, for which we observed no threat, although some areas are well conserved while others are not.

**Table 1.** Presence of the different habitats in each SCI (A, B, C, D, E). A= SCI Almaden, Chillon and Guadalmez, B= SCI Sierra the Canalizos, C= SCI Sierra Morena, D= SCI Rivers of the middle Guadiana basin and slopes, E= SCI Lagunas de Ruidera.

| HABITAT | A | B | C | D | E |
|---|---|---|---|---|---|
| 4030 | | x | x | x | |
| 5210 | | | x | | x |
| 5330 | x | x | x | x | x |
| 6310 | x | x | x | x | x |
| 9230 | | x | x | x | |
| 9240 | | x | x | x | x |
| 9320 | x | x | x | x | |
| 9330 | x | x | x | x | |
| 9340 | x | x | x | x | x |
| 9540 | | | x | | |
| 9561 | | | | | x |

Table 1 shows the differences in the type and number of habitats, in which C (Sierra Morena) has the highest number of habitats, but very similar to the SCIs A, B, D; however, all these sites, which are characterized by their siliceous substrates, are totally different from E (Lagunas de Ruidera), whose habitats are located on calcareous substrates, which accounts for their high degree of floristic difference.

## 5. Conclusions

As a result of the study of five SCIs in the central Iberian Peninsula, we present maps of the plant associations of *Quercus* at a scale of 1:10,000, and characterize the different associations from the botanical–ecological point of view for their inclusion in their corresponding habitats. These act as the fundamental basis for the conservation of other EU priority habitats. The use of these woods is therefore possible and even desirable, as their transformation into wooded pastures is of vital importance for livestock and for $CO_2$ fixation [41–44]. In the case of the cork oak woods, we observed a poorer conservation status. Although these woods have been used for cork extraction, we believe it is necessary to implement forestry management actions in order to rejuvenate the trees. Unlike the holm oak and cork oak woods, special protection measures should be considered for Portuguese and Pyrenean oak woods; the Portuguese oak woods of *Q. canariensis* and *Q. marianica* are scarce and very fragile formations, so they require special measures. This is also the case of *Q. pyrenaica* woods, as these latter occur on palaeosoils; both *Q. canariensis* and *Q. pyrenaica* are located particularly in the central rainier areas, where there is also a high number of EU priority habitats. There are wooded formations of *Juniperus*, belonging to habitats with special protection, which must be mapped and their conservation studied to ensure their protection [34,36].

The areas in this study are of particular importance and have been mapped as a result of the general guidelines imposed by the European Commission on the member states. This is the reason that a large number of national parks, nature reserves, and SCIs in Spain have been mapped at a scale of 1:10,000 in order to incorporate them in the Natura 2000 network. All the mapped areas have been considered previously by the EU as areas of community interest for protection.

**Author Contributions:** Conceptualization: A.C.-O., C.J.P.G., and E.C.; data curation: A.C.-O., C.M.M., R.Q.C., J.C.P.F., S.d.R., J.M.H.I., G.S., C.J.P.G., and E.C.; formal analysis: E.C.; investigation: A.C.-O. and E.C.; methodology: A.C.-O., C.M.M., and E.C.; project administration: E.C.; resources: E.C.; supervision: A.C.-O., C.M.M., and E.C.; validation, A.C.-O., C.M.M., J.C.P.F., S.d.R., C.J.P.G., and E.C.; software: J.C.P.F. and S.d.R.; visualization: A.C.-O., C.M.M., R.Q.C., and E.C.; writing—original draft: A.C.-O. and E.C.; writing—review and editing: A.C.-O., C.M.M., R.Q.C., J.C.P.F., S.d.R., C.J.P.G., and E.C. All authors have read and agreed to the published version of the manuscript.

**Funding:** This work does not have funding for its publication and is based on the research project: "Scientific collaboration agreement for the knowledge of the vegetation, the types of habitats and the distribution of the threatened flora in spaces of the Natura 2000 Network in the province of Ciudad Real between the public company Environmental Management of Castilla La Mancha SA (GEACAM SA) and the University of Jaén".

**Institutional Review Board Statement:** Not applicable.

**Informed Consent Statement:** Not applicable.

**Acknowledgments:** We would like to thank the Government of Castilla-La Mancha, Spain, the company GEACAM, and the University of Jaén for their assistance during this research.

**Conflicts of Interest:** The authors declare no conflict of interest.

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
