# Peer review of "Forest and Arborescent Scrub Habitats of Special Interest for SCIs in Central Spain"

_land, doi:10.3390/land10020183_

Round 1

Reviewer 1 Report

this is an interesting and important contribution.  I suggest defining SIC/LIC in the paper and the abstract so readers unfamiliar with EU jargon can understand.

My main reservation, communicated to the Editor, is that I am not sure Land is the right Journal.  I suspect Phytocenologia or Journal of Vegetation Science would garner a greater readership and the paper have thus more influence.

Author Response

Thank you for your suggestions, we add a paragraph in the summary briefly explaining the meaning of places of community interest

Reviewer 2 Report

The manuscript needs strongly improvements in the form and in the contents. I din not find the novel contribution of the research, given that the manuscript is presented as a report more than a scientific paper. The captions are not clear and the results are a long list of maps without a description.

Author Response

Thanks for your suggestions, we have added a paragraph at the end of the introduction explaining the novelty of the study.

We add comments in the results section, specifically in each figure.

Syntaxon codes have been revised

The taxon is authored when naming it for the first time

The linguistic corrections have been reviewed by a native translator

Reviewer 3 Report

The authors propose an article entitled “Habitats of forests and arborescent scrub with special interest for the SCIs of central Spain”. The manuscript is original in the conceptual idea, well structured, with new data relating to habitats from directive 92/43 EEC, with particular attention to oak forests. The conservation of these habitats are of considerable importance for the conservation of biodiversity in Europe, so I think the work deserves to be published. In the text I correct some few grammar mistake and I suggest few corrections for a better presentation of the work.

Author Response

Thank you for your suggestions, we have taken your suggestions into account, words and phrases that you have suggested have been eliminated, we have reviewed all the italics, and all taxa have been authored when they are named for the first time. We've checked the codes for the syntaxons. We explain in material and methods the lack of code xxxxxx.

The linguistic corrections have been reviewed by a native translator

Round 2

Reviewer 1 Report

The authors have made a number of improvements and with a final check of the English it is suitable for publication.  

Author Response

It has again revised the English by a native specialist person

Thanks for your comments

Reviewer 2 Report

The manuscript is mainly a botanical/phytosociological study focused on specific habitats with the the incorporation of the new syntaxa described prior to the implementation of the EU code in five sites of community interest (SCIs). The study is not well presented and it is too much focused on a specific case study.

Introduction: this section is missing of the background literature about the SCI, the role of their monitoring, which are the indicators used to monitor their conservation status. 

Materials and Methods: the citations after GIS (19,20, and 21) means that the implementation of these data within a GIS has been already published in these references?

Results:

3.1. Cartographic analysis of the main habitats of forests and arborescent shrubs (maps) 

The maps are not adequately presented and described, the contents of the maps are not in English language, the captions of the figures are not clear.

Discussion

Table 3 should be better included in the text and discussed

Conclusions

They need to be improved. Which is the relevance of the research, also other EU countries should do the same research? why?

Author Response

19, 20,21. They are quotes from where the mapped syntaxons were obtained.

Information about the color of the polygons is placed at the bottom of the figures to clarify the information on the map.

Table 3 does not exist, I imagine it refers to table 1, this table expresses the different habitats present in the SCIs

The conclusions have been improved, see text

It has again revised the English by a native specialist person

Thanks for your comments
